# Deep Learning Segmentation in 2D echocardiography using the CAMUS dataset : Automatic Assessment of the Anatomical Shape Validity

**Sarah Leclerc** [1]                                       SARAH.LECLERC@CREATIS.INSA-LYON.FR
**Erik Smistad** [2]                                             ERIK.SMISTAD@NTNU.NO
**Andreas Ostvik** [2]                                          ANDREAS.OSTVIK@NTNU.NO
**Frederic Cervenansky** [1]                        FREDERIC.CERVENANSKY@CREATIS.INSA-LYON.F
**Florian Espinosa** [5]                                     FLORIAN.ESPINOSA@GMAIL.COM
**Torvald Espeland** [2,3]                                 TORVALD.ESPELAND@NTNU.NO
**Erik Andreas Rye Berg** [2,3]                             ERIK.A.BERG@NTNU.NO
**Pierre-Marc Jodoin** [4]                           PIERRE-MARC.JODOIN@USHERBROOKE.CA
**Thomas Grenier** [1]                               THOMAS.GRENIER@CREATIS.INSA-LYON.FR
**Carole Lartizien** [1]                             CAROLE.LATIZIEN@CREATIS.INSA-LYON.FR
**Lasse Lovstakken** [2]                                   LASSE.LOVSTAKKEN@NTNU.NO
**Olivier Bernard** [1]                             OLIVIER.BERNARD@CREATIS.INSA-LYON.FR

[1]*CREATIS, CNRS UMR5220, Inserm U1044, INSA-Lyon, Villeurbanne, France.*

[2]*Center of Innovative Ultrasound Solutions (CIUS), Department of Circulation and Medical Imaging, Norwegian University of Science and Technology (NTNU), Trondheim, Norway*

[3] *Clinic of cardiology, St. Olavs Hospital, Trondheim, Norway* [4] *Computer Science Department, University of Sherbrooke, Sherbrooke, Canada*

[5] *Cardiovascular department, Centre Hospitalier de Saint-Etienne, Saint-Etienne, France*

## Abstract

We recently published a deep learning study on the potential of encoder-decoder networks for the segmentation of the 2D CAMUS ultrasound dataset. We propose in this abstract an extension of the evaluation criteria to anatomical assessment, as traditional geometric and clinical metrics in cardiac segmentation do not take into account the anatomical correctness of the predicted shapes. The completed study sheds a new light on the ranking of models.

**Keywords:** Cardiac segmentation, deep learning, ultrasound, left ventricle, myocardium

## 1. Introduction

The segmentation of ultrasound images of the heart has long been done with semi-automatic methods due to the lack of robustness of automatic algorithms (Armstrong et al., 2015). With the recent advances in supervised deep learning, it is now possible to achieve the inter-expert accuracy with the right quantity and quality of data. In our recent paper (Leclerc et al., in press), we used the now publicly available CAMUS dataset (500 patients acquired in both two and four chamber views - AP4C - AP2C) to analyze the potential and the behavior of deep learning methods on multi-structure cardiac segmentation.

## 2. Deep Learning for Segmentation using an Open Large-Scale Dataset in 2D Echocardiography

In (Leclerc et al., in press), we performed extensive evaluation of encoder-decoder architectures, focusing on the U-Net design, which outperforms state-of-the-art methods with an accuracy that is within the inter-expert variability.

### 2.1. CAMUS dataset

We set up a dataset (CAMUS) that includes 1000 2D ultrasound sequences (2 chamber and 4 chamber views of 500 patients) along with the reference contours of three structures (the left ventricle -LV, the myocardium and the left atrium) at the end diastolic and end sysolic instants, annotated by one cardiologist expert. Inter-expert and intra-expert variabilities were computed on one of ten folds (50 patients).

### 2.2. Evaluated algorithms

Eight state-of-the-art algorithms were compared including non-deep learning methods (Structured Random Forests -SRF (Dollar and Zitnick, 2015), B-Spline Active Surface Models - BEASM, fully or semi-automatic (Pedrosa et al., 2017)), and encoder-decoder networks (two U-Nets (Ronneberger et al., 2015) -U-Net 1 and U-Net 2, Stacked Hourglass -SHG (Newell et al., 2016), U-Net ++ (Zhou et al., 2018) and Anatomically Constrained Neural Networks -ACNNs(Oktay et al., 2018)). Each algorithm went through a ten-fold cross-validation.

### 2.3. Main conclusions

Focusing on the U-Net architecture, we showed that :

- U-net is robust to image quality, and able to segment several structures without dropping performance at any instant (ED and ES) or view (AP4C and AP2C). It also generalizes well with only a 250 patients training set (half of the dataset), while still benefiting from additional training data, unlike SRF (Leclerc et al., 2018).
- Encoder-decoder networks outperform state-of-the-art non-deep learning methods and its accuracy is within the inter-expert scores. However, they still do not meet intra-observer scores and still produce a significant amount of geometrical outliers ( 18%).
- Deep convolutional neural networks with a more complex architecture do not outperform the U-Net, neither on geometrical metrics nor clinical indices. This implies that the U-Net design is sufficient to learn the task complexity.

## 3. Anatomical metrics based on shape analysis of expert segmentations

### 3.1. Shape Simplicity and Convexity

Inspired by a recent work published on natural images (Zhu et al., 2017) to compare the segmentation of several structures S by different annotators, we present two geometrical criteria that maintain characteristic values on successfully segmented cardiac structures.

$$Convexity : Cx(S) = \frac{Area(S)}{Area(ConvHull(S))} \quad \text{and} \quad Simplicity : Sp(S) = \frac{\sqrt{4\pi * Area(S)}}{Perimeter(S)}$$

These two metrics give discriminative values for any convex shapes, such as the oval-like shape of the LV and the bridge-like shape of the myocardium. Therefore, we could derive from expert scores thresholds to separate properly segmented images from cases where models produced anatomically impossible shapes, which we called anatomical outliers.

### 3.2. Application to Camus

We provide in table 1 the geometric scores of the experts and of U-Net 1 and U-Net 2 on the inner (LV-endo) and outer contours (LV-epi). The minimum values from the experts' annotations are used as thresholds to label segmentations as anatomical outliers. Both models show on average lower scores on both structures compared to the experts.

Table 1: Additional geometric scores and outlier rates computed on the full dataset. *ana : anatomical , geo : geometrical*

| Method | # Trainable Parameters | LV-endo | | LV-epi | | Outliers | | |
|---|---|---|---|---|---|---|---|---|
| | | **Cx** | **Sp** | **Cx** | **Sp** | **geo** | **ana** | **geo ∩ ana** |
| Experts | − | 0.975 ±0.022 >0.741 | 0.722 ±0.040 >0.529 | 0.992 ±0.004 >0.960 | 0.794 ±0.022 >0.694 | − | − | − |
| U-Net 1 | 2M | **0.958** ±0.022 | **0.665** ±0.037 | **0.976** ±0.012 | **0.743** ±0.022 | 423 21% | 95 ±5% | 71 ±4% |
| U-Net 2 | 18M | 0.952 ±0.030 | 0.658 ±0.045 | 0.970 ±0.028 | 0.732 ±0.045 | 519 26% | 318 ±16% | 231 ±12% |

Though U-Net 2 outperformed U-Net 1 on the Dice, Mean Absolute Distance (MD) and Hausdorff Distance (HD) (Leclerc et al., in press), it produces three times less anatomically plausible shapes, possibly because of its higher number of parameters. The derived criteria are sensitive to any local deformity, as illustrated in yellow in Fig. 1. These results confirm that traditional metrics are not sufficient to rank algorithms, in particular learning methods.

### 4. Conclusion

We open the door for a more appropriate evaluation of segmentation results in 2D echocardiography via the introduction of anatomical metrics that complete our original study.

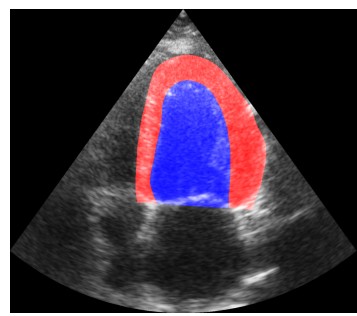 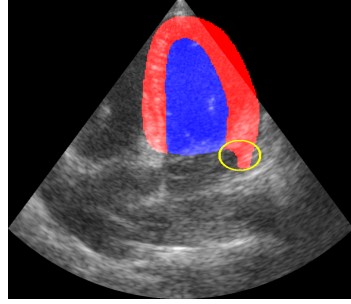 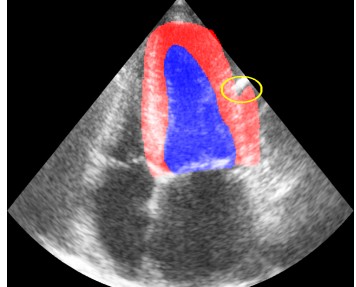

(a) Expert segmentation
Cx= 0.99|0.99, Sp= 0.78|0.80

(b) Ana outlier 2CH-ED
Cx= 0.96|0.94 Sp= 0.67|0.71
M= 1.3|2.8 HD= 5.0|14.1 mm

(c) Ana outlier 4CH-ES
Cx= 0.93|0.95 Sp= 0.65|0.70
MD= 2.1|2.1 HD= 5.5|6.4 mm

Figure 1: Ana outliers (different patients) : b) is also a geometrical outlier but not c).

## Acknowledgments

This work was performed within the framework of the LABEX PRIMES (ANR- 11-LABX-0063) of Université de Lyon, within the program "Investissements d'Avenir" (ANR-11-IDEX-0007) operated by the French National Research Agency (ANR).
The Center for Innovative Ultrasound Solutions (CIUS) is funded by the Norwegian Research Council (project code 237887).

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
