# OpenReview forum: "Deep Learning Segmentation in 2D echocardiography using the CAMUS dataset : Automatic Assessment of the Anatomical Shape Validity"
_MIDL.io/2019/Conference/Abstract — MIDL Abstract 2019_

### Official Review · AnonReviewer1 · 2019-04-30
**Additional measures based on geometry of segmented structures are presented for better comparison of methods**

**Rating:** 3
**Confidence:** 2

**Review:**


Summary:

Additional metrics based on anatomical geometry of segmented cardiac structures are presented to complete a comparative study presented in a related work by the authors. Using shape convexity and simplicity as additional measures, it is claimed that segmentation methods can be ranked more appropriately.

Comments:

+ Use of measures beyond voxel level accuracy like dice overlap is a useful idea. At the end of the day, dice or voxel related measures are surrogate metrics and use of relevant anatomical geometry measures can be suitable in several applications.
+ As the work is based on an extended work, the experiments are comprehensive

- It is not well motivated as to why shape complexity measures are better suited for this application.
- The paper is not self-contained and one has to read Leclerc et al. to fully understand this work. For instance, nowhere the differences between U-Net 1 and U-Net 2 are explained. Applies to several other discussion points.

---

### Official Review · AnonReviewer2 · 2019-04-30
**Summary abstract of relevant journal submission**

**Rating:** 3
**Confidence:** 2

**Review:**

The authors summarize the key findings from a recent journal paper submission, comparing state-of-the-art models for semantic segmentation and demonstrating near expert-level performance on multi-label cardiac segmentation from echocardiography images. The authors also release the largest publicly available echocardiography dataset of 4-chamber and 2-chamber image planes.

The abstract submission additionally proposes the use of two additional geometric criteria from (Zhu et al., 2017), convexity and simplicity, which they propose help identify anatomical outliers. They demonstrate how a  U-net with much greater capacity, while producing best results on more established metrics including Dice coefficient and Hausdorff Distance, produces more geometric abnormalities as compared to a U-net with a smaller capacity, and these abnormalities can largely be identified with the new metrics.

The proposed criteria are simplistic, but appear to be suitable for the identification of displayed geometric abnormalities.

The paper is clearly presented, and the referenced journal paper submission along with the proposed geometric criteria will certainly be of interest to the medical imaging community.

Minor comment:

- Section 2: "auto-encoder architectures" should be "encoder-decoder architectures" - the target output is not the same as the input.

Section 3.2: "difformity" should be "deformity"

---

### Decision · Program_Chairs · 2019-05-06
**Acceptance Decision**

Accept